# Effects of Chemical State of the Pd Species on H_2_ Sensing Characteristics of PdO_x_/SnO_2_ Based Chemiresistive Sensors

**DOI:** 10.3390/s19143131

**Published:** 2019-07-16

**Authors:** Tianjiao Qi, Jie Sun, Xi Yang, Fanfan Yan, Ji Zuo

**Affiliations:** Institute of Chemical Materials, China Academy of Engineering Physics, Mianyang 621999, China

**Keywords:** hydrogen sensor, low temperature, SnO_2_, chemical state of Pd

## Abstract

In this paper, the PdO_x_ nanoparticles modified SnO_2_ are prepared using sputtering and wet chemical methods. The SnO_2_ nanoparticles are separately added to a concentration of 0.75% to 10% PdCl_2_ to obtain a PdCl_2_/SnO_2_ composite material, which is calcined for 1 to 2 h at the temperatures of 120 °C, 250 °C, 450 °C and 600 °C. The PdO_x_/SnO_2_ nanocomposite was characterized by X-ray photoelectron spectroscopy (XPS), X-ray diffractometry (XRD) and transmission electron microscopy (TEM). Microstructural observations revealed PdO_x_ with different chemical states attached to the surface of SnO_2_. Hydrogen response change tests were performed on the obtained PdO_x_/SnO_2_ gas sensing materials. The results show that the high gas sensing performance may be attributed to the contribution of the PdO_x_-loaded SnO_2_. In hydrogen, the best sensitivity response was attained at 80 °C, which is 60 times that of pristine SnO_2_. It clarifies the role of PdO_x_ in the gas sensing mechanisms.

## 1. Introduction

Hydrogen is expected to become a green and renewable energy source, in response to air pollution, global warming and the increasing shortage of fossil fuels. However, this light and odorless gas is highly flammable, and the leakage can result in disastrous consequences, such as explosions [1]. Therefore, there has been a huge demand for effective hydrogen sensors that can be used for providing a warning about the leakage of hydrogen during the production, storage, delivery, and usage stages [2].

Metal oxide semiconductors based chemiresistive gas sensors are widely used to detect varieties of inflammable gases and toxic pollutants [3,4], due to their high sensitivity, simplicity in fabrication, low cost, and simple measuring electronics [5]. As a typical n-type wide bandgap semiconductor, tin oxide (SnO_2_, with a bandgap (E_g_) of 3.6 eV at 300 K) has been highly investigated due to its high gas sensitive activity to a wide variety of gases [6]. Meanwhile, their poor selectivity creates a huge limitation for achieving wide applications. Furthermore, the high working temperature (usually superior to 200 °C) requires high power consumption, and restricts the integration and the use of materials for device assembling [7].

In order to improve the H_2_-selectivity and decrease in operating temperature of SnO_2_-based gas sensors, surface modifications of SnO_2_ with PdO_x_ nanoparticles, have been widely employed [8,9]. The electronic sensitization mechanism has been proposed to explain the improvement in gas-sensing performance [10,11]. The reaction with H_2_ molecules takes place on the surfaces of PdO_x_ nanoparticles and not on the SnO_2_ [12]. The PdO_x_ nanoparticles change their charge state, which induces a variation of the surface barrier height and therefore leads to a conductance change on the SnO_2_ [13]. In this case, the PdO_x_ nanoparticles act as a “receptor” through a specific interaction with H_2_ molecules [14], and SnO_2_ only has a “transducer” role of the changes induced by the interaction of PdO_x_ nanoparticles with H_2_ molecules [15].

The PdO_x_ nanoparticles modified SnO_2_ can be prepared though sputtering and wet chemical methods [16,17]. However, the impregnation-calcination method has been extensively used to prepare PdO_x_/SnO_2_ composites using PdCl_2_ as precursor due to its technological simplicity. In spite of numerous papers reporting on the improvements in the H_2_-sensing performances by using PdCl_2_ as a precursor, the origin of these improvements [18] and the correlation of H_2_-sensing characteristics and chemical state of the introduced materials haven’t been investigated systematically [19]. 

In this study, we studied the sensing behavior of the PdO_x_/ SnO_2_ sensor at different annealing temperatures that obtained different chemical states of Pd toward H_2_ at the optimal operating temperature. From the study, the different chemical states of Pd substantially participated in the gas sensing reaction, thereby improving the H_2_ sensing performance of the sensor.

## 2. Materials and Methods

### 2.1. Preparation of PdO_x_/SnO_2_ Composites

The PdO_x_/SnO_2_ composite was obtained using a simple synthesis-calcination method. First, 5 mol% of PdCl_2_ was dissolved in ethanol, then SnO_2_ nanoparticles having a particle diameter of 50 to 70 nm were added to a PdCl_2_/ethanol solution and thoroughly mixed, sonicated for 30 min. The prepared mixture was dried at 100 °C to obtain PdCl_2_/SnO_2_ composites. Finally, the PdCl_2_/SnO_2_ composite was calcined at a heating rate of 5 °C min^−1^ for 2 h at 120 °C, 250 °C, 450 °C, and 600 °C under an air atmosphere to obtain a final sample. According to the difference in calcination temperature, the obtained PdO_x_/SnO_2_ composites were abbreviated as PdO_x_/SnO_2_-120, PdO_x_/SnO_2_-250, PdO_x_/SnO_2_-450, and PdO_x_/SnO_2_-600, respectively.

### 2.2. Characterization

To examine the chemical states of components sensitivity materials, all the sensors were characterized by X-ray photoelectron spectroscopy (XPS, VG 250, Thermo); the powder phases were analyzed by X-ray diffraction (XRD, Bruker D8) with Cu-Kα radiation (λ = 1.5406 Å). The morphology was analyzed using field emission scanning electron microscopy (SEM) and transmission electron microscopy (TEM). The SEM was performed on a FEI Sirion-200 with an acceleration voltage of 10 kV. The TEM was carried on a Tecnai G2F20 with a typical acceleration voltage of 200 kV. 

### 2.3. Fabrication and Measurement of Gas Sensor

The gas-sensing characteristics of H_2_ in different chemical state Pd-doped SnO_2_ based sensors were characterized. The as-synthesized composites were mixed with terpineol, and then, the pastes were uniformly coated on the Au-alloy electrodes and dried at 100 °C to obtain the gas sensors. The schematic diagram and photograph of Au-alloy electrode are shown in Figure 1a, which is a composite of an alumina substrate (area = 1 mm × 1.5 mm), two Au electrodes on the upper surface, and a micro heater on the lower surface. The photograph of the final gas sensor is shown in Figure 1b. The gas sensors were aged to make sure the sensitivity was stable before the first measurement.

The gas-sensing characteristics of the gas sensors were operated on a specially designed home-made gas sensing analysis system. The electrical current of sensor is measured by an electrochemistry analyzer (CHI1000B, Chenhua Shanghai) with a potential of 1 V. The heating voltage is supplied by a power voltage supplier. A constant flux of air of 100 sccm as gas carrier was mixed with the gas to be measured and dispersed in the synthesized air to obtain different gas concentrations. All measurements were conducted in a temperature-stabilized sealed chamber. Air was injected in until the system became stable, then the system switched to alternating between the test gas and carrier gas every 15 min. The gas flow rates were precisely manipulated using a computer controlled multi-channel mass flow controllers, and humidity was controlled under 40 RH%.

The sensing response was defined as: S = I_g_/I_a_, where the I_a_ is the background current in carrier gas, and I_g_ is the changed current in test gas.

## 3. Results and Discussion

### 3.1. Microstructure Characterization of PdOx/SnO_2_ Composites

The XRD patterns of SnO_2_, PdCl_2_/SnO_2_, and PdO_x_/SnO_2_ composites were indicated in Figure 2. The recognizable diffraction peaks can be assigned to the tetragonal SnO_2_ (JCPDS 41-1445). There is no recognizable peak of PdO_x_ or other crystalline phases appeared, probably due to the low crystallization and small content of the additives.

The crystalline sizes were measured by means of an X-ray line-broadening method, using the Scherrer equation: D = kλ/β_hkl_cos θ(1)

D is the crystalline size in nanometers; λ is the wavelength of the radiation (1.54056Å for CuKα radiation); k is 0.94; β_hkl_ is the peak width at half-maximum intensity, and θ is the peak position.

It can be seen that the peaks of Pd doped SnO_2_ NPs with different annealing temperature are wider than the peak of no annealing. The crystallite size calculated by the Scherrer method is decreasing with annealing temperature increasing. In 1991, Yamazoe demonstrated that a reduction in crystal size would significantly increase the sensor’s performance [20]. 

The microstructure of PdO_x_/SnO_2_-450 composite was investigated by TEM and STEM-EDS characterizations. The TEM image of PdO_x_/SnO_2_-450 is shown in Figure 3, where the size of nanoparticles is 50 to 70 nm and the shape is a sphere. The particle size is consistent with that of SnO_2_. Furthermore, the rough surface morphology of PdO_x_/SnO_2_-450 composite is obviously different from the smooth surface morphology of SnO_2_ nanoparticles, indicating that the PdO_x_ nanoparticles are homogenously dispersed on the surfaces of SnO_2_ nanoparticles.

The corresponding elemental mapping of Sn, O, and Pd reveals that the PdO_x_ nanoparticles are uniformly dispersed on the surfaces of SnO_2_ nanoparticles.

### 3.2. Chemical States of PdO_x_ Dispersed on the Surfaces of SnO_2_

X-ray Photoelectron Spectroscope proves a very effective method in identifying the elements of the materials, chemical states and electronic states of the elements [21]. 

In Figure 4a–d, we plotted evolution of the Pd 3d_5/2_ spectrum in the dependence on the different annealing temperature. The Pd1s spectra of the XPS results were analyzed with more details. During annealing treatment, three characteristic peaks Cl-Pd-Cl (BE = 337.3 eV), Pd-O (BE = 336.6 eV), and O-Pd-O (BE = 338.1 eV) are observed and these peaks indicate the changes of chemical state of palladium. The asymmetric peak was belonged to two components with BE of 337.3 eV and 336.6 eV, with spectrums that are typical of palladium in the oxidation state of Pd^2+^ and Pd^4+^. And increasing the temperature to 450 °C resulted in the peak Pd (BE = 337.3 eV) which corresponds to the Cl-Pd-Cl structure having disappeared in the spectra. When the temperature is to 600 °C, we also observed further changes that the binding energy (BE) of Pd was shifted to lower values for smaller particles, see Figure 4a, and the intensity ratio of peak O-Pd-O to peak Pd-O increased gradually. According to Table 1, it is interesting in the series of spectra in Figure 4a–d that pure Pd^0^ or Pd^n+^ situations are never obtained. 

For more understanding of the chemical states of Pd, the O 1s spectra of the XPS results were analyzed with more details in Figure 4e–h. The asymmetric O 1s peak was deconvoluted into three components with binding energies of O-Sn-O (BE = 530.9 eV), Pd-O (BE = 531.3 eV) and O-Pd-O (BE = 532.3 eV) [23]. As the temperature increases to 120 °C, two new peaks BE = 530.9 eV and BE = 531.3 eV of O-Sn-O, and Pd-O, appear respectively. As the temperature reaches 250 °C, a peak BE = 532.3 eV is observed to be the structure of O-Pd-O. When the temperature is in the range of 250 °C to 600 °C, the intensity ratio of peak O-Pd-O and Pd-O is gradually increased. Figure 4 illustrates the process of Pd surface change on Pd-SnO_2_. O_2_ diffuses to the surface of Pd and adsorbs on the surface of Pd to form an active adsorption state. Then, the oxygen atoms and Pd atoms on the surface of Pd undergo an orderly remodeling to form PdO. As the temperature increases, the oxygen atoms accumulated on the surface further react with PdO to form PdO_2_. The chemical state transition of palladium and the order in which these peaks appear are as follows:$${\mathrm{PdCl}}_{2}\overset{120\;{}^{\circ}C}{\underset{\begin{array}{c} \mathrm{partial}\\ \mathrm{oxidation} \end{array}}{\to }}{\mathrm{PdCl}}_{2}/\mathrm{PdO}\overset{250\;{}^{\circ}C}{\underset{\begin{array}{c} \mathrm{further}\\ \mathrm{oxidation} \end{array}}{\to }}{\mathrm{PdCl}}_{2}/\mathrm{PdO}/{\mathrm{PdO}}_{2}\overset{\geq 450\;{}^{\circ}C}{\underset{\begin{array}{c} \mathrm{more}\\ \mathrm{oxidation} \end{array}}{\to }}\mathrm{PdO}/{\mathrm{PdO}}_{2}$$

Figure 5 shows the variation of the Sn 3d (a) binding energy (BE) of the PdCl_2_/SnO_2_ nanoparticles with the different annealing temperature. As shown in Figure 4b, the two BEs have the same tendency, which indicates that the BE of Sn 3d_3/2_ and O 1s decreases with increasing temperature, the value of Sn is 1.92 eV, the value of O is 1.82 eV, and the oxygen in SnO_2_ loses electrons. As a result, the Fermi level shifts and bends upward, forming a depletion layer.

### 3.3. H_2_ Sensing Characteristics of PdO_x_/SnO_2_ Composites

To determine the optimum operating temperature, we tested the response of Pd-doped SnO_2_ gas sensors at different operating temperatures. The response of the material to 100 ppm H_2_ was tested at 25, 50, 80 and 100 °C and the results are shown in Figure 6. As the operating temperature increases, the response of the Pd-doped SnO_2_ first increases and reaches a maximum at 80 °C, whereas the temperature continues to increase, and the sensitivity is reduced. This is caused by the gas adsorption/desorption process on the sensor surface of Pd-doped SnO_2_. When the working temperature rises to a certain temperature, the desorption rate of the gas increases, resulting in a decrease in sensitivity. At different annealing temperatures, there is a similar trend.

Normally, the reactivity between the target gas and adsorption oxygen needed certain activation energy, which was provided by increasing the work temperature. Since the Pd nanoparticles have a catalytic effect, the activation energy is lowered, thus the optimum working temperature is lowered to 80 °C. As the temperature increases, the diffusion rate of gas molecules increases, and it is easier to react with gas-sensitive materials. In addition, at high temperatures, due to a large decrease in adsorbed oxygen, the grain boundary barrier decreases and the conductance increases, so that the resistance change decreases when contacted with H_2_. The gas sensitivity performance is declining. In the experiments, the following sensing characteristic of Pd-doping SnO_2_ was tested at the optimal operating temperature (80 °C). 

The sensor is annealed from 120 °C to 600 °C, showing a high corresponding sensitivity. The results of four reproducibility tests performed at the optimum operating temperature are shown in Figure 7, indicating that the sensor has good repeatability and stability.

The response recovery time is also one of the important parameters for evaluating the gas sensor, as shown in Figure 8. Gas sensing materials are often required to have fast response and recovery characteristics in applications. We can see that the response time is similar as 100 s at PdO_x_/SnO_2_-120, -250, -450, and the response time at PdO_x_/SnO_2_-600 is slower than the others. The recovery time is faster at higher temperature than those in lower temperature (from 120 °C to 450 °C). The recovery times were 340 s and 185 s for the sensor based on annealing temperature at 120 °C and 250 °C, respectively, and 140 s for the sensor based on annealing temperature at 450 °C. Moreover, the recovery time of annealing temperature at 600 °C is 290 s that is not better than lower temperature. It can be seen that when hydrogen is introduced, the resistance of the sensor is reduced and then the steady state is reached. The nanostructures provide a larger specific surface area that can speed up the adsorption and desorption processes, which shorten response and recovery time.

Figure 8 also demonstrates that the sensitivity of the PdO_x_/SnO_2_ at the different annealing temperature. The sensitivity of sensor at 450 °C is higher than the others. These features are produced by the influence of grain boundaries between the nanoparticles. In XRD, we know that the crystallite size is decreasing as the annealing temperature is increasing. By studying the effect of grain size on the sensitivity of the H_2_ sensor, it is found that the larger the particle size of the nano-gas sensitive material, the lower the sensitivity.

Figure 9a shows that the response of PdO_x_/SnO_2_ to different concentrations of H_2_ in air. The sensor chips were tested at an optimum temperature of 80 °C It can be found that the sensor show strong responses from 2.5 ppm to 100 ppm. Figure 9 shows that as the H_2_ concentration increases, the response sensitivity increases. When exposed to H_2_, the mechanism is that hydrogen converted to atomic hydrogen at the material interface, causing the current signal to change.

The corresponding sensitivity can be obtained from the slope of the standard curve and the gas concentration. This result indicates that the sensitivity is related to the adsorption of H_2_ molecules on the surface of the gas sensing material.

The response stability of the PdO_x_ modified SnO_2_ gas sensor to 100 ppm H_2_ was tested at 80 °C with 15 cycles (Figure 9c). As the number of times increases, the sensitivity decreases and gradually reaches a steady state. This may be due to the deposition of some contaminants on the sensor surface after the first few cycles. Obviously, Pd decorated SnO_2_ has good stability and can be used in various practical applications. 

We investigated the gas response characteristics of the sensor to different concentrations of PdO_x_ loaded SnO_2_ at its optimum operating temperature. Sensors of 0.75 wt%, 1.25 wt%, 2.5 wt%, 5 wt%, and 7.5 wt% PdCl_2_ supported SnO_2_ were prepared and annealed at 450 °C. The results are shown in Figure 10, where the change in sensitivity indicates that the corresponding performance is highly correlated with the Pd loading amount. We can see that the 5.0% by weight load sensor more sensitive to hydrogen than other concentrations of PdCl_2_. Because of more PdO_x_ is on the surface of SnO_2_, the stronger reaction captures electrons and generates an electron depletion layer on the surface of SnO_2_. The higher loading of Pd leads to its full coverage on the surface of SnO_2_, which reduces the contact between gas and SnO_2_. The effect of PdO_x_ greatly exceeds the influence of SnO_2_ bulk material, so that the performance of SnO_2_ itself can’t be fully reflected, and therefore the sensitivity decreases.

Figure 11 shows the experimental results of the effect of humidity on the H_2_ sensitivity. We have performed sensor response tests of Pd/SnO_2_-450 under relative humidity from 20% to 70% conditions. The sensitivity changed significantly in different humid environments. It can be showed that Pd/SnO_2_-450 has less sensitivity in low and high relative humidity at an operating temperature of 80 °C, while the corresponding sensitivity to 100 ppm H_2_ is highest at 40% humidity. This is that water vapor increases the electron concentration of Pd/SnO_2_-450, thereby increasing oxygen absorption. When the relative humidity is further increased, the sensitivity starts to decrease, which is caused by the synergistic competition between H_2_ and water vapor.

The selectivity of the gas sensor is another important parameter for assessing the sensing capabilities of gas sensing materials. The gas selection performance of the material was obtained by testing the sensitivity of PdO_x_-doped SnO_2_ to 100 ppm different kinds of gases. The testing gases include hydrogen (H_2_), ethanol (EtOH), carbon monoxide (CO) and methane (CH_4_). As can be seen from Figure 12, the Pd-doped SnO_2_ gas sensor exhibits significantly higher response sensitivity to H_2_ and a smaller change to other gases. Therefore, we believe that the gas sensor has good selectivity for H_2_.

### 3.4. Sensing Mechanism

The sensing mechanism of semiconductor material sensors is based on semiconductor oxygen vacancy reactions. In air, due to the difference in temperature, molecular oxygen may exist in the form of O^2−^, O^−^ or O_2_^2−^, etc., and oxygen forms an electron depletion layer by depleting electrons on the surface of the material to lower the electrical conductivity. When the palladium atom acts as a catalyst, its function is to dissociate oxygen on the surface of SnO_2_, thus increasing the overflow of oxygen from the surface of SnO_2_ to obtain a greater amount of electrons and thereby increasing sensor sensitivity [24,25,26]. In addition, in order to increase the conductivity, the palladium atom can be divided into two processes. First, hydrogen molecules dissociate into hydrogen atoms on the surface of Pd, and then hydrogen atoms adsorbed at moderate temperatures act as donors, providing additional electrons to the material and inducing a cumulative layer. 

With increasing the annealing temperature, different chemical states of Pd doped SnO_2_ complex materials were obtained (Figure 13). The Pd^2+^ additive on SnO_2_ substrate is reduced to a Pd metal, which plays an important sensitivity role in hydrogen and leads to change of conductivity. But partial PdCl_2_ of the sensor on a low annealing temperature was hydrolyzed to Pd(OH)_2_ in wet air and it is generation of [PdCl_4_]^2−^ in different pH. From the potential of the reduction reaction, we know that Pd(OH)_2_ and [PdCl_4_]^2−^ are more difficult to reduce to catalytic Pd metal than Pd^2+^. So the sensitivity of the sensors at a low annealing temperature is worse than the high annealing temperature.
Pd^2+^ + 2e→Pdφ = 0.915 V.Pd(OH)_2_ + 2e→Pd + 2OH^−^φ = 0.07 V[PdCl_4_]^2−^ + 2e→Pd+ 4Cl^−^φ = 0.621 V
$$\mathrm{Pd}+H_{2}\Leftrightarrow {\mathrm{PdH}}_{x}$$

It is deduced that Pd^4+^ is the key factor to promote the sensitivity. Basing on the valence electron layer structure of platinum group metals, the stability of high-valent Pd(Ⅳ) is lower than Pd(Ⅱ). So the conversion of Pd^4+^ is easier. On the high annealing temperature, the proportion of PdO_2_ is increasing. This is one of the reasons why sensor chips at high annealing temperatures have better gas sensitivity. In addition, while increasing the annealing temperature, the sensitivity performance was stronger because of the unpreventable decrease of the grain sizes. In our study, when the annealing temperature is higher than 600 °C, the gas sensitivity is reduced in contrast. The detailed mechanism of different Pd chemical states doping SnO_2_ affecting the performance of the sensor still needs further investigation. 

## 4. Conclusions

In conclusion, SnO_2_ nanoparticles loaded with PdO_x_ at different annealing temperatures have been systemically studied for H_2_-sensing applications. XPS and XRD were used to study elements analysis and crystalline sizes changes that accompany the different chemical states phase transformation. The sensor chips were produced by coating with as-preparation Pd-doped SnO_2_ materials, with preformed interdigitated planar heatertype sensor electrodes. In particular, the different annealing temperature Pd loaded SnO_2_ film had a good stability and exhibited the best H2-sensing performances, which were increased by a factor of 15 at an optimum operating temperature of 80 °C. In addition, different annealing temperature Pd/SnO_2_ complex materials show much higher H_2_ selectivity against CO, CH_4_ and CH_3_CH_2_OH. Compared with annealing temperature at 120 °C, 250 °C, 450 °C and 600 °C, the 450 °C showed enhanced H_2_ sensing properties, such as a higher response (260–100 ppm H_2_), rapid response/recovery time, and good selectivity and stability. The sensing behavior of the sensor can be understood simply by the potential of the reduction reaction and the oxygen ionosorption model. PdO and PdO_2_ play important roles in modifying SnO_2_ by means of H_2_ adsorption on the sensor chips. 

## Figures and Tables

**Figure 1 sensors-19-03131-f001:**
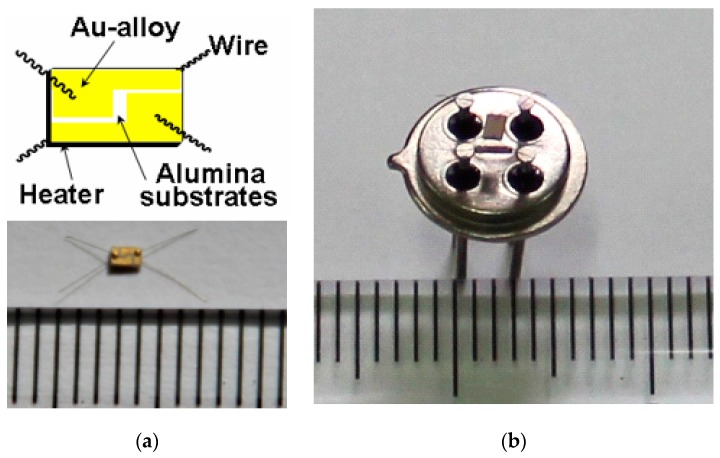
(**a**) Schematic diagram and photograph of Au-alloy electrode, which is composite of an alumina substrate (area = 1 mm × 1.5 mm), two Au electrodes on the upper surface, and a micro heater on the lower surface. (**b**) Photograph of the final gas sensor.

**Figure 2 sensors-19-03131-f002:**
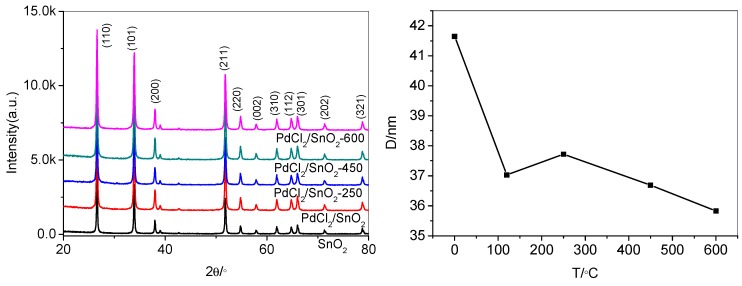
XRD patterns and grain size of SnO_2_, PdCl_2_/SnO_2_, and PdO_x_/SnO_2_ composites.

**Figure 3 sensors-19-03131-f003:**
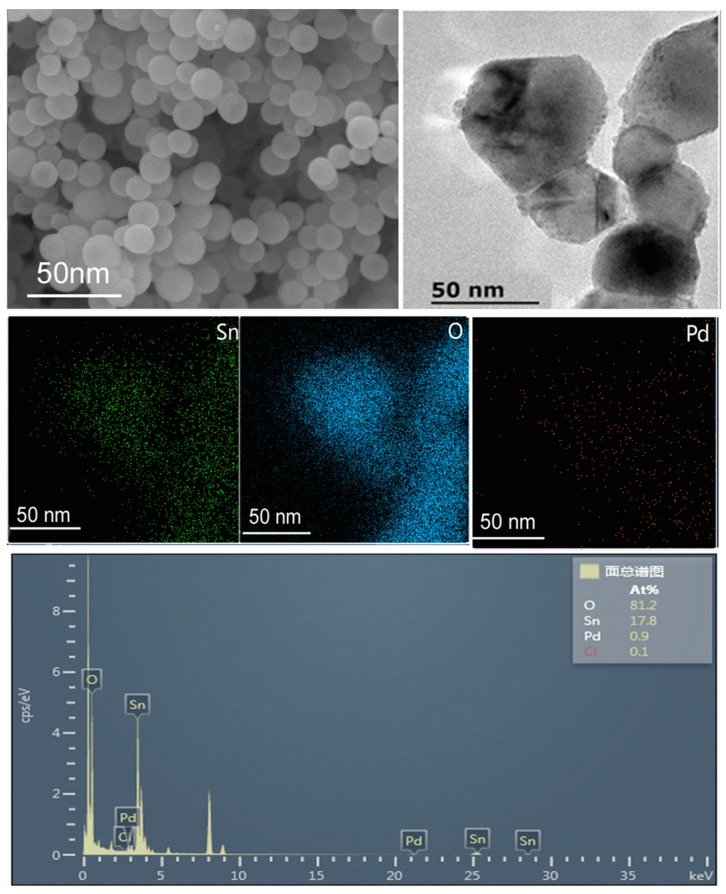
Bright-field TEM image of PdO_x_/SnO_2_-450 composite and the corresponding STEM-EDS elemental maps.

**Figure 4 sensors-19-03131-f004:**
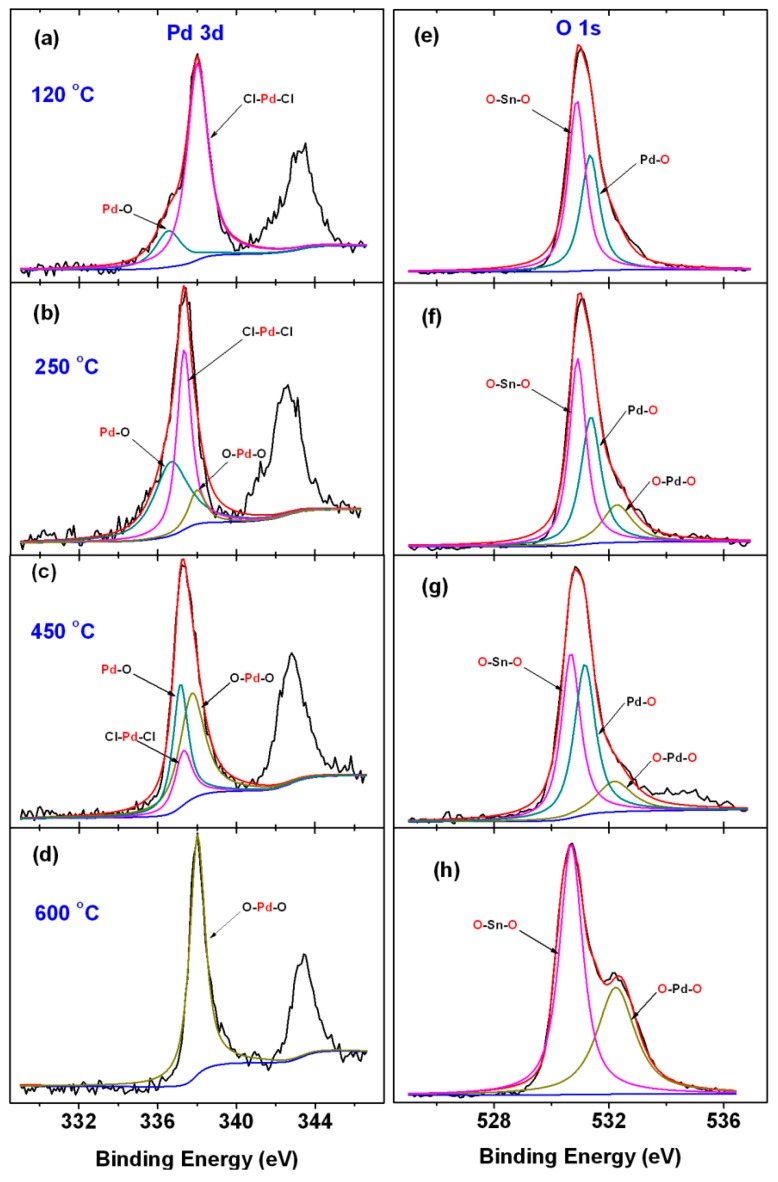
XPS narrow scan spectra (**a–d**) the Pd 3d XPS spectra; (**e–h**) the O 1s XPS spectra.

**Figure 5 sensors-19-03131-f005:**
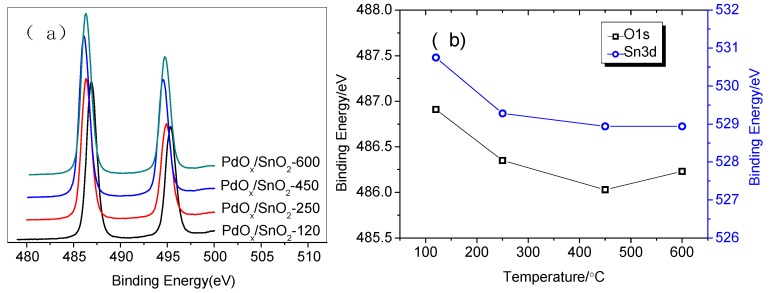
(**a**) The typical Sn 3d XPS (**b**) The change trend that the BE of Sn 3d3/2 and O 1s.

**Figure 6 sensors-19-03131-f006:**
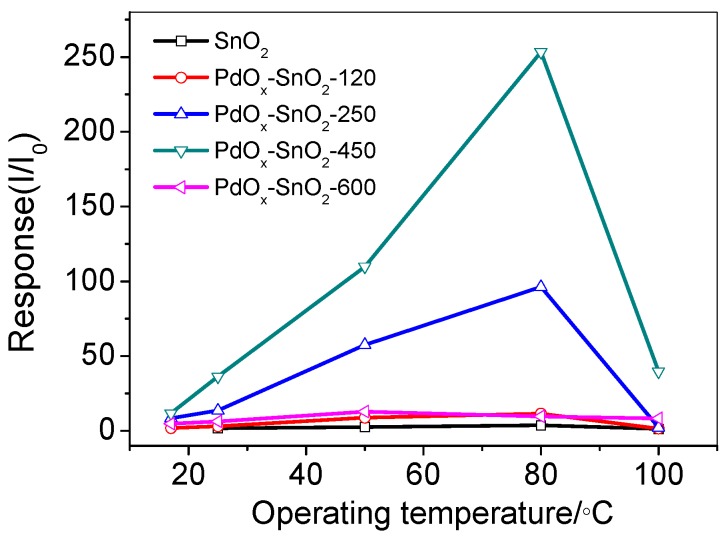
The sensors responses to 100 ppm H_2_ at PdO_x_/SnO_2_ composites.

**Figure 7 sensors-19-03131-f007:**
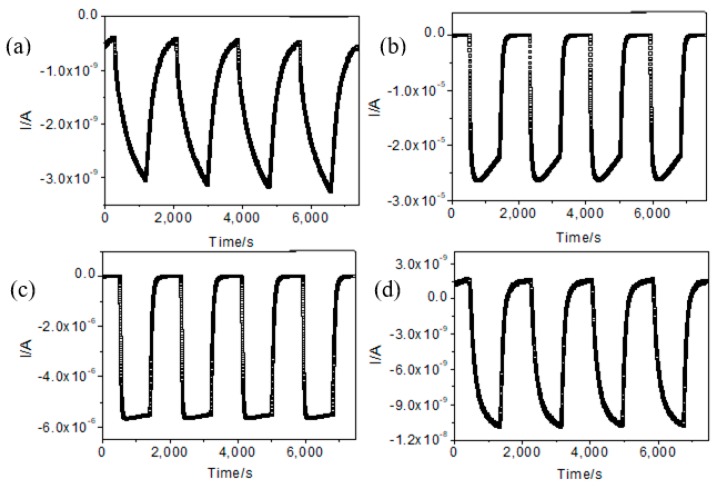
Four periods of response curve of the sensor using different annealing temperature at their optimal operating conditions respectively at 100 ppm H_2_: (**a**) PdO_x_/SnO_2_-120; (**b**) PdO_x_/SnO_2_-250; (**c**) PdO_x_/SnO_2_-450; (**d**) PdO_x_/SnO_2_-600.

**Figure 8 sensors-19-03131-f008:**
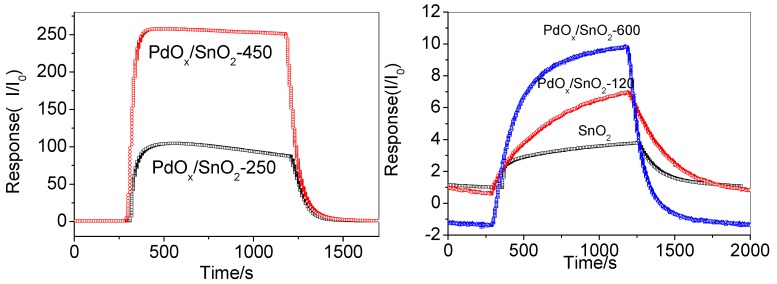
Response and recovery characteristics of PdO_x_/SnO_2_ composites to100 ppm H_2_.

**Figure 9 sensors-19-03131-f009:**
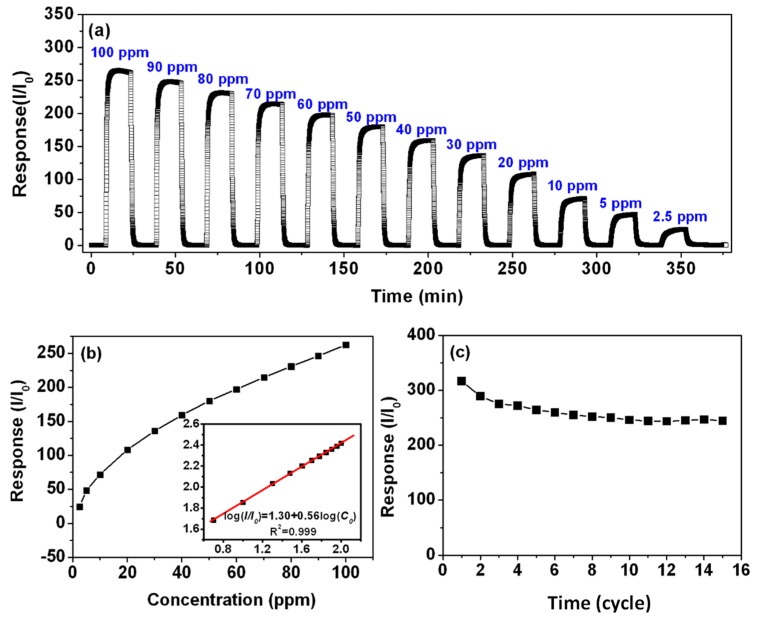
(**a**) Response (S) of sensor based on PdO_x_/SnO_2_ to various concentration of H_2_ at optimal operating temperature and (**b**) linear dependence of response (S) of H_2_ gas sensor based on PdO_x_/SnO_2_ on concentration of H_2_ gas at optimal operating temperature, (**c**) the response stability of the PdO_x_ modified SnO_2_ gas sensor to 100 ppm H_2_ with 15 cycles.

**Figure 10 sensors-19-03131-f010:**
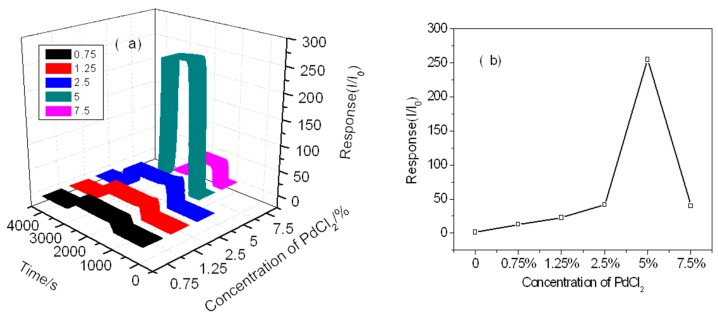
Different concentrations of PdCl_2_ loaded at theirs optimal operating temperatures respectively to 100 ppm H_2_: (**a**) Response transients of the sensor from 0.75% to 7.5% Pd loaded, (**b**) Sensor response as a function of Pd concentration.

**Figure 11 sensors-19-03131-f011:**
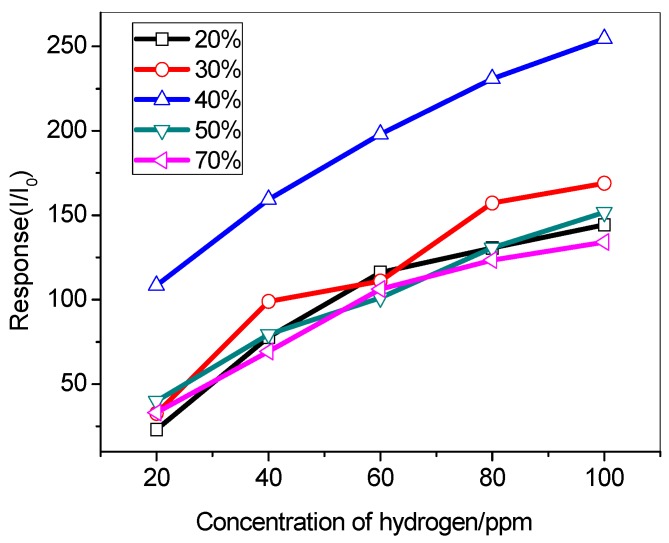
Responses of the Pd/SnO2-450 °C to different concentration of hydrogen at 20%–70% relative humidity.

**Figure 12 sensors-19-03131-f012:**
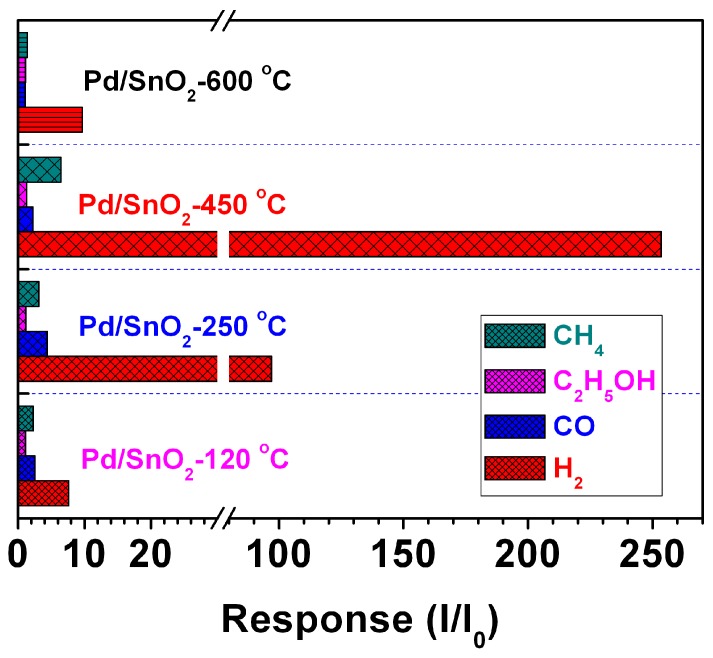
Responses of the PdO_x_/SnO_2_ to 100 ppm different gases at their optimum operating temperature.

**Figure 13 sensors-19-03131-f013:**
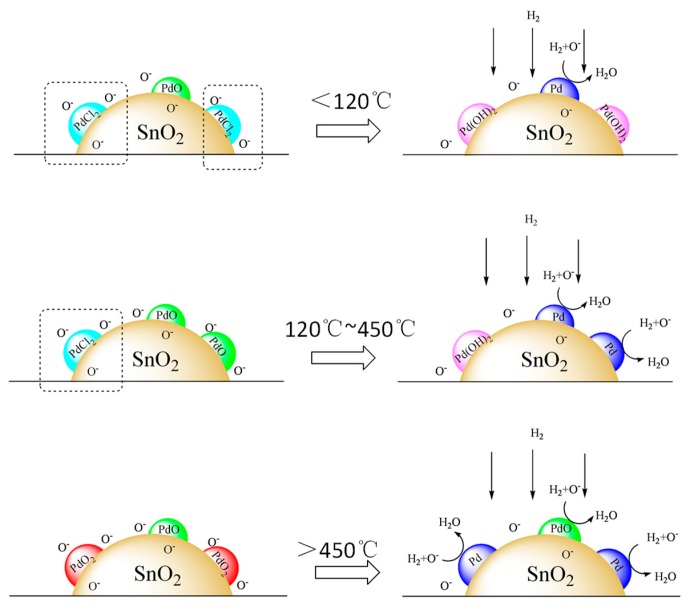
Schematic representation of different chemical states of Pd on the SnO_2_, as well as the proposed mechanism of H_2_ sensitivity.

**Table 1 sensors-19-03131-t001:** BE values for Pd 3d_5/2_ photoemission signal in several reference compounds [22].

Compound Bulk	Pd 3d_5/2_ BE (eV)
Pd	334.8–335.1
PdO_x_,x < 1	335.3
Pd-O	336.9
O-Pd-O	337.9
Cl-Pd-Cl	337.8

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
