# Peer review of "Effects of Chemical State of the Pd Species on H2 Sensing Characteristics of PdOx/SnO2 Based Chemiresistive Sensors"

_sensors, 2019, doi:10.3390/s19143131_

Round 1

Reviewer 1 Report

Tianjian et al present a study aiming to relate the chemical state of Pd to the hydrogen sensing performance of PdOx/SnO2 sensors. They achieved different Pd states by annealing the precursors, comprising of PdCl2/SnO2, at different temperatures. While the goal of the study in of high interest for the Pd-based hydrogen sensing community, the execution (and related discussion) of the study is very poor. The main problem is how unfocused the manuscript is. As title suggests, the readers expect to find a clear study design relating the chemical states and sensing performance. Instead, the experimental design goes everywhere, for example, discussing grain boundaries. The discussion of results is also held at very shallow level; only describing what is seen at the figures. Why some trends happens? Why there are exceptions to them? The manuscript is also plagued by grammatical problems that make reading uncomfortable. I suggest to get assistance with professional proofreader to improve the readability. Hence, at current state, I cannot recommend publication. Below I have several points that I'd like the author to address:

  Reference 1, 2, 5 and 7 discuss general statement, however the cited paper are specific work on hydrogen sensors. This is not at all relevant. In general the reference are really poor. Reference 27 is invoked in line 112, while the reference list only has 24 papers. How come??  

Line 51 mentioned PdCl2/SnO2. I understand that this is the precursor. However, to not confuse readers, better directly refer them as PdO/SnO2.

Figure 2. Please add the peak reference for the elements. Also, please plot the extracted crystallite size. Describing them in words is not enough for readers.

Line 110. How is it possible that higher annealing temperature produces smaller crystallite size? Higher annealing should drive the system towards single crystallinity i.e. bigger crystalline.

Line 118. Use sphere instead of ball.

Line 120. Authors mentioned smooth surface for SnO2 particles. This has to be shown.

Line 128-138. This discussion on the Pd chemical states as function of annealing should be the center of the manuscript. Yet this is very poorly discussed. I had very hard time trying to get useful information from this paragraph. Related figure 4 is also really poor. Readers cannot get useful information quickly. Line 136: what smaller particles?

Line 158-163. Here no discussion on why different annealing temperature (i.e. chemical state) gives rise to different signal. Also, why signal at 100C is out of trend? At least give some rationalization based on available literature.

Line 166-168. This explanation is baseless. 

Line 188-191. How do the authors know that it is the surface area that define the speed? They dont characterize this in the manuscript. Also, bigger crystallite leads to slower response/recovery time.

Line 195-197. Can the author rationalize this statement?

Figure 10. Why higher loading of Pd give lower signal?

Reviewer 2 Report

The manuscript reported the effects of Pd ionic species of PdOx/SnO2 hybrid on H2 sensing performances. The mechanism study is precisely done with material characterization (XPS, EDS) and sensing performance. However, the present version of manuscript doesn’t present a good quality for the publication. Based on this, I suggest reconsider this paper for publication after address the comment and the following concerns: 

1.      What is experimental evidence for sentence 117-118? In addition, how can authors say surface morphology of pure SnO2 nanoparticles are smooth? Authors should provide morphological characterizations of pure SnO2.

2.      Figure quality of EDS graph in Figure 3 is very low. Please plot the graph using raw data, not paste the image file.

3.      Graph format should be identical through whole manuscript. For example, Figure 5a and 5b format is very different even in same figure making readers to be confused.

4.      What is reason for the description that PdO2 is dominant in high temperature?

5.      In Figure 7, readers can be confused with that temperature in graph is indicating which one: oxidation temperature or operating temperature?          

6.      Authors should do sensing performance comparison between pure SnO2 and PdOx/SnO2 with various species.

7.      In Figure 13, authors should indicate which scheme indicating which samples.

8.      Some related references regarding to hydrogen sensing mechanism of Pd should be cited such as: ACS sens. 2018, 3, 1876-1883

9.      Reference form should be identical (e.g. Ref. 18, 22)

Round 2

Reviewer 1 Report

Authors have modified the manuscript according to comments in the previous review round. In general all concerns are addressed and thus the manuscript is improved. I recommend publication ONLY after professional proofreading is done to the text. I have suggested this in the previous review, which was entirely neglected by the authors.

This is just a couple of examples, from the abstract and conclusion:

 "Microstructural observations revealed that the different chemical states PdOx nanoparticles attached to the surface of SnO2." Should be: Microstructural observations revealed PdOx with different chemical states attached to the surface of SnO2.

"In particular, the different annealing temperature Pd loaded SnO2 film had a good stability exhibits the best H2-sensing performances with 15 times at optimal operating temperature of 80 ℃." I dont even understand the sentence here.

There are a lot of similar language errors in the manuscript. Please take this advise seriously.

Author Response

revised in manuscript